# Intelligent Medical System with Low-Cost Wearable Monitoring Devices to Measure Basic Vital Signals of Admitted Patients

**DOI:** 10.3390/mi12080918

**Published:** 2021-07-31

**Authors:** Siraporn Sakphrom, Thunyawat Limpiti, Krit Funsian, Srawouth Chandhaket, Rina Haiges, Kamon Thinsurat

**Affiliations:** 1School of Engineering and Technology, Walailak University, Nakhon Si Thammarat 80160, Thailand; siraporn.sa@wu.ac.th (S.S.); krit.fu@wu.ac.th (K.F.); csarawou@wu.ac.th (S.C.); 2Center of Excellence on Wood and Biomaterials, Walailak University, Nakhon Si Thammarat 80160, Thailand; 3Center of Excellence for Sustainable Disaster Management, Walailak University, Nakhon Si Thammarat 80160, Thailand; 4National Institute of Public Administration (INTAN), Kuala Lumpur 50480, Malaysia; rina@intanbk.intan.my

**Keywords:** wireless body area network (WBAN), low-cost wireless sensor devices, internet of things (IoT), intelligent medical system

## Abstract

This article presents the design of a low-cost Wireless Body Sensor Network (WBSN) for monitoring vital signs including a low-cost smart wristwatch that contains an ESP-32 microcontroller and three sensors: heart rate (HR), blood pressure (BP) and body temperature (BT), and an Internet of Things (IoT) platform. The vital signs data are processed and displayed on an OLED screen of the patient’s wristwatch and sent the data over a wireless connection (Wi-Fi) and a Cloud Thing Board system, to store and manage the data in a data center. The data can be analyzed and notified to medical staff when abnormal signals are received from the sensors based on a set parameters from specialists. The proposed low-cost system can be used in a wide range of applications including field hospitals for asymptotic or mild-condition COVID-19 patients as the system can be used to screen those patients out of symptomatic patients who require more costly facilities in a hospital with considerably low expense and installation time, also suitable for bedridden patients, palliative care patients, etc. Testing experiments of a 60-person sample size showed an acceptable accuracy level compared with standard devices when testing with 60 patient-samples with the mean errors heart rate of 1.22%, systolic blood pressure of 1.39%, diastolic blood pressure of 1.01%, and body temperature of 0.13%. According to testing results with 10 smart devices connected with the platform, the time delay caused by the distance between smart devices and the router is 10 s each round with the longest outdoor distance of 200 m. As there is a short-time delay, it does not affect the working ability of the smart system. It is still making the proposed system be able to show patient’s status and function in emergency cases.

## 1. Introduction

With the pandemic situation of Coronavirus Disease 2019 (COVID-19) currently happening globally, as of 28 April 2021, more than 146 million people have been infected resulting in the number of 3 million deaths, roughly 2.12% of the total infected cases [1]. In the first wave of the pandemic, confirmed cases were reported mostly in developed countries such as the USA and European countries where medical facilities are fully supported; however, because of a massive number of infected cases, some patients were not properly treated as necessary facilities such as an intensive care unit (ICU), a mechanical ventilator, etc., are limited. At the moment, most infected COVID-19 cases are reported in countries where medical facilities are limited with high population numbers such as India, Southeast Asia countries, South America countries, etc., which may lead to a higher number of cases/fatalities ratio in the near future if additional medical facilities are not provided in time.

The most common symptom at the onset of critical COVID-19 patients are cough and fever [2,3], while patients with mild-COVID-19 usually have the symptoms of cough, hyposmia, and sputum, mostly without fever symptoms [4]. Therefore, having a device that can measure the body temperature of confirmed COVID-19 cases integrated with intelligent health monitoring systems may reduce medical staff’ workload and risk of getting infected from a regular screening procedure for the initial symptoms of severe symptomatic patients from mild COVID-19 or asymptomatic patients in field hospitals resulting in higher efficiency of resources management in pandemic situations [5]. Although there are commercial healthcare monitoring systems available, the price of a system, including the necessary devices, is extremely high, which is not suitable to be integrated in field hospitals in some areas of the world. Therefore, a low-cost healthcare monitoring system used to monitor vital signals of admitted patients was proposed in this research to demonstrate required equipment, design procedure for integrated systems, and software development, not only for the essential benefit to fight through the COVID-19 pandemic situation but also for future healthcare systems for monitoring admitted patients in hospitals to reduce the regular checks workload of medical staff with an intelligent alert system proposed in this research.

Intelligent biomedical systems and healthcare technologies have been developed and deployed in several medical applications which contribute to longer life expectancy of populations in many countries where there is no financial deficit for the medical systems [6]. For example, Seoane et al. [7] developed a real-time mental stress monitoring system using multiple wearable biomedical devices to extend the operational capability of Spanish combatants. The finding was that cardiac and respiratory signals are preferred for stress assessment. Zheng et al. [8] reviewed various works in the field of biomedical systems that used piezoelectric and triboelectric energy generators to harvest energy in a human body from internal motions for powering biomedical sensors in a body that contributes to extending lifespan and enhancing the quality of life of patients. Hinchet and Kim [9] also addressed the importance of self-power biomedical devices to facilitate future autonomous biomedical systems with the internet of things. Successful biomedical systems should be reliable in terms of accuracy, adaptable with local environments, easy to use by medical staff, and comfortable with patients.

In addition to advanced development of biomedical devices, a healthcare monitoring system is a crucial part to reducing regular workload of medical staff with patients’ consolation to know that they are real-time monitored during hospitalization. Care monitoring devices are used to measure vital signs of critical patients usually attached, wired or wireless, on parts of a patient’s body. Discharged patients can also equipped with a home-monitoring system to send health data to hospital. A care monitoring system consists of two main parts which are a wearable system and low-power circuits and sensors. A wearable system is an unharmed device compatible with the human body—for example, artificial nerves used to imitate the human auditory system by using an auditory brainstem which enhance users’ hearing ability and cochlear implants’ performance [10]. Another example is a wireless neurosurgery device controlled by the near-field inductive coupling operation between the implantation system and the external system. It has the advantage of a small surgical wound with low energy usage for the system when a high data transmission rate between the implantation to the external system is required [11]. Normally, not a single sensor but multi-sensors are used to simultaneously monitor patients’ condition for surgical operation or even the diagnosis process. Therefore, data transmission technology must be able to handle multiple data transfer from sensors with high data rate without data loss [12,13,14].

Wireless Body Sensor Network (WBSN) is related to a Wireless Body Area Network (WBAN) and has been used extensively nowadays. The WBAN is a necessary technology for a real-time medical checkup that influences researchers to develop the WBSN to measure vital signals of a patient’s body such as heart rate, glucose level, blood pressure, muscle, and motion. Multi-sensors have been developed and connected to a wireless sensor network system as shown in Figure 1. Sensors are attached on a human body to measure several signals such as electrocardiogram (ECG), electromyography (EMG), electroencephalography (EEG), blood pressure, respiratory, and movement activity [15]. Signals are then sent from the sensors to sending and receiving nodes that are connected to a controller or a personal server. A short-range radio frequency is used to send data from the receiving node to the personal server used to store all medical signals data from sensors. The stored data can be processed and displayed to patients. The processed data can also be sent to a medical server via WLANs (2G, GPRS, 3G) [14,15,16,17]. A medical server is used to record, review, organize patients’ data, and perform medical diagnosis to be able to give advice in an emergency situation. The diagnosis results will be reviewed by a doctor again and, for normal cases, patients do not have to see a doctor in person [16], which can significantly reduce medical staff’ workload.

In the time of the COVID-19 pandemic situation, the demand for field hospitals is dramatically increased, but the amount of medical staff may not be included in field hospitals with full capacity due to a limited amount of staff. Therefore, a healthcare monitoring system integrated with necessary sensors to monitor patients’ vital signs is in great demand; nevertheless, all sensors and systems should be inexpensive, easy to use, and commercially available to be able to fight through the pandemic situation.

Although there are advanced sensors and reliable healthcare monitoring systems being used in leading hospitals around the world such as in [18], those systems are expensive and there is limited knowledge published in detail of how to construct the entire system, which may widen the gap for people to get an equal level of medical treatment. Al Bassam et al. [19] presented a useful IoT system with a wearable device to monitor the signs of quarantined remote patients of COVID-19; however, blood pressure, which is one of the vital signs, is not included. Wan et al. [20] developed a wearable IoT real-time health monitoring system to monitor blood pressure, heart rate, and body temperature, where sensors are specifically designed for the vital signals, and the MySQL database was used, which is quite expensive compared to an open-source database. Ru et al. [21] presented a detailed IoT monitoring system to monitor human health with high accuracy sensors with a sample size of three people. Marathe et al. [22] used an ECG module along with a pulse oximeter, blood pressure, and temperature sensors to monitor 11 patients, but the algorithm to classify the patients’ symptom level was not included. Chowdary et al. [23] used Raspberry PI3 to process pulse, temperature, blood pressure, and fingerprint signals to monitor health as a prototype system while Akram et al. used Arduino for monitoring patient health; however, they were limited details and validations to be used in this pandemic situation.

This research presents the methodology to design an inexpensive intelligent medical system with small and low-cost sensors to measure basic vital signals which are pulse, heart rate, blood pressure, and body temperature with high accuracy and the ability to recognize the working environment for 60-person sample-size. Data of the sensors will be sent and displayed on a screen attached to the patient’s body. Patients’ data will also be able to be displayed in a mobile application for their relatives if permitted. Moreover, patients’ data will be stored in the hospital’s data center for medical staff to verify, analyze, and use as an initial diagnosis process to classify patients into three levels which are normal, surveillance, and risky levels before contacting the doctor for further treatments or detail diagnosis. The proposed system, which is inexpensive and easy to be reproduced without specialists, can apply not only in the COVID-19 field hospital but also for regular hospitals on newborns, the elderly, bedridden patients, etc.

### Organization of the Paper

This research divides into five sections including introduction of the application of an intelligent medical system followed by sensors used in the WBSN that described the methodology for measuring blood pressure, pulse, and heart rate. The third section is the proposed architecture systems, which is the design concept of smart devices for measuring vital signs and how to connect with the data center shown on the display screen of the patient’s wristwatch using the IoT platform. In the fourth section, the validation method is described followed by the Results section, which shows testing results with the sample size of 60 persons. Finally, the Conclusions section presents the final results and suggests an intelligent medical system with low-cost wearable monitoring devices for smart healthcare that can be applied to several applications, including the COVID-19 field hospital.

## 2. Sensors Used in the WBSN for Vital Signs

### 2.1. Blood Pressure, Pulse, and Heart Rate

Blood pressure (BP) is the pressure in the artery caused by pumping pressure from the heart to circulate blood throughout the body via a circulatory system. A blood pressure can be measured with two numbers, which are the systolic and diastolic blood pressures. Systolic blood pressure (SBP) is the maximum pressure in blood vessels when the ventricles are contracting. Diastolic blood pressure (DBP) is the minimum pressure in blood vessels when the ventricles are relaxing. Normal blood pressure is between 120/80–139/89 mm of mercury (mm Hg).

Arterial pulse is the rhythmic dilation of the artery sent to the ventilatory system after leaving the heart. It causes the arterial pulsation, which is called the pulse. It has the speed of wave pulses around 5–8 m/s, which is 10–15 times faster than the speed of blood flow in the arteries. The pulse can be measured from a radial artery because it is easy to detect and can be measured in the unit of beat per minute (BPM). Generally, the pulse and heart rates are equal if the heart compression is sufficiently strong; otherwise, the pulse rate will be less than the heart rate. Heart rate and pulse rate are the vital signs besides blood pressure, respiratory rate, and body temperature used to classify patients into normal and abnormal stages.

Normal pulse rate: The normal pulse rate, or heart rate in the normal condition, is 60–80 bpm after 10 min of resting for adults. Pulse rate depends on factors such as age (children ages 6–15 around 70–100 beats per minute, adults around 60–100 beats per minute)), resting (slower pulse rate), emotion, depression, anxiety (faster of pulse rate), medication, athlete (the slower of pulse rate), and diseases such as thyroid toxemia (faster pulse rate). Normal pulse rate of adults is on average 70–72 bpm, 60–70 bpm for men, and 72–80 bpm for women. The pulse rate of women is higher than men because of the smaller size of the heart as women require a lower volume of blood and lower hemoglobin level than men.

Abnormal pulse rate: There are three types of abnormal pulse rate: fast heart rate, which is faster than 100 bpm, slow heart rate, which is slower than 60 bpm, and arrhythmia, which means that the heart may beat faster or slower than normal rates or with an irregular rhythm. Normally, there are no symptoms for abnormal pulse rate, but they may feel faint or dizzy while the heart is abnormally beating [24].

### 2.2. Method of the Measuring Pulse

There are two major techniques to measure the pulse: Electro-cardiogram (ECG) and Photoplethysmography (PPG). ECG directly measures the electrical signal produced from the heart in each pulse. The devices using an ECG technique have to be attached on a patients’ chest, which may not be comfortable. Using a light technology to measure the change in blood volume due to the cardiac cycle is called the Photoplethysmography (PPG) technique. PPG detects the change of hemoglobin absorption by using reflection or transmission properties of light in the body’s tissues, which is a non-harmful technique resulting in being more comfortable for patients and ease of use for medical staff to monitor a cardiovascular system. Because of the mentioned advantages of the PPG technique, it is generally preferred for measuring the pulse for most patients.

The PPG technique with light transmission property can be performed on any thin body parts such as fingertips. A light transmitter (light emitting diode or infrared diode) and a light receiver are gripped in opposite sides of a finger as shown in Figure 2a. The wavelength can penetrate through hemoglobin and tissues in the finger to the receiver which can be photodiode, LDR, and phototransistor. The blood volume at the fingertip changes according to heart contraction and expansion which can be used to measure the pulse. At the time of heart contraction of a cardiac cycle, more blood circulates into vessels; thus, the transmitter sends less light to the receiver. Conversely, when there is a relaxation phase of the cardiac cycle, the concentration of the red blood cell decreases, resulting in more light being able to pass through a finger to the receiver. The intensity of light being received by the receiver is converted into the form of an electrical signal and then amplified for ease of observation. Noise may be included in the signal that have to be filtered before being further processed in a microcontroller to obtain the signal in the unit of beats per minute (BPM), the number of times your heart beats in one minute.

A device gripping a fingertip may make it difficult for a patient to complete regular activities; therefore, a PPG with a light reflection property is developed as a wearable device, such as a wristband or watch. The transmitter and receiver are aligned with each other as shown in Figure 2b, which is more convenient and flexible for patients. However, it is likely to move from the preferred wrist position, which may lead to a less accurate pulse measurement. Therefore, if the pulse signal of patients is a vital sign, using the light transmission property to monitor the pulse is preferable for most applications.

### 2.3. Relationship between Blood Pressure and Heart Rate

Blood pressure (BP) is the kind of force that the heart performs to pump blood to all body parts. Systemic Vascular Resistance (SVR) is the resistance between the arteries’ wall and the blood, and a human body normally has a certain level of SVR. In irregular conditions, the narrower arteries result in higher resistance, which leads to high BP. The arteries expanding or the patient being dehydrated may result in lower SVR, which is the cause of low blood pressure. Cardiac Output (CO), known as heart output, can be calculated from CO = SV × HR, where SV (Stroke Volume) is the blood volume pumping out of the heart in a beat of contraction and HR (heart rate) is the number of heartbeats per minute. With all parameters as aforementioned, blood pressure can be calculated from Equations (1) and (2):(1)BP=CO×SVR
(2)BP=SV×HR×SVR

According to Equations (1) and (2), blood pressure directly varies corresponding to the stroke volume that is pumped out of the heart, and this reflects the amount of blood in a body. If the body is bleeding or dehydrated, the volume of blood decreases, which makes the heart beat faster to deliver the normal demand of the body. In the case of lipid deposition in the arterial wall or the lack of flexibility, the artery cannot expand to adjust the resistance value, which contributes to high blood pressure.

### 2.4. The Concept Idea of Vital Sign Monitoring Devices

A low-cost and uncomplicated design of vital sign monitoring devices mostly applies the principles of light as presented in Section 2.1, Section 2.2 and Section 2.3, and to examine the heart rate/pulse rate and blood pressure of a patient. All signals are sent via an ESP8266 microcontroller which sends the data via Wi-Fi. Optionally, the data can be sent via an ESP32, able to send the data via both Wi-Fi and Bluetooth. Selection of a data transmission method depends on the distance between sensors and a display screen. Bluetooth is used for exchanging data over short distances while Wi-Fi is able to exchange data over long distances. Therefore, wireless transmission may reduce the workload of doctors and nurses for rechecking the vital signs of patients as data enter the gateway and display in real-time on monitor screens or an application in a smartphone for authorized medical staff who are in charge of a patient. In this work, all data were sent via Wi-Fi.

## 3. The Proposed Overall Architecture System

This research presents wireless sensor devices integrated with a health monitoring system to monitor vital signs of internal patients in hospitals. Normally, nurses are required to regularly check vital signs of internal patients every 3–4 h which may physically and mentally disturb patients and their relatives while resting. Therefore, a real-time monitoring device is designed to monitor vital signs without bothering the patients. The proposed architecture of the intelligent medical system for measuring body vital signs in this paper is divided into two parts: (1) designing of a smart device for a Wireless Body Area Sensor Network (WBSN) and (2) designing of an intelligent medical monitoring system.

### 3.1. Designing of a Smart Device for a Wireless Body Area Sensor Network (WBSN)

A smart device for a Wireless Body Area Sensor Network (WBSN) is designed to measure vital signs, which are heart rate, pulse, blood pressure, and body temperature. The concept is that the device should have no side effects when attached to a human body. It should be small so that it is comfortable for patients like a smart watch or a wristband. Moreover, it has to be inexpensive compared with other devices commercially available so that countries with limited budgets can use them in field hospitals to deal with the COVID-19 pandemic situation and also use them in normal hospitals to reduce medical staff workload. The design is based on the structure of wireless sensor nodes composed of sensing units used to measure vital signs before being processed by a microcontroller. Both transmitter and receiver send/receive data through RF waves. The vital signs’ data, in a so-called data patient’s room, are processed and displayed on a patient’s wristwatch and on a patient’s relative mobile phone with a patient’ consent. The data are also sent as a star network to the central database of the hospital via a wireless network to be displayed on an IoT platform for medical staff as shown in Figure 3. Figure 3 shows a structure that consists of wireless body sensors connected to the ESP32-Microcontroller, a small and portable wireless module, able to transmit data via Wi-Fi and Bluetooth; however, all data were sent via Wi-fi for this work. The transmission of each sensor is sent as a star network from the ESP32′s serves as a sensor node to the other ESP32+Wi-Fi Microcontroller that serves as the master to collect and display the data on a computer or a mobile phone. Vital signs data from each bed are sent to a healthcare gateway as presented in Figure 4.

### 3.2. Design of the Intelligent Medical Monitoring System

Wireless Communication for an intelligent medical system is communication between a database in a patient’s room and a healthcare database center. It compares and displays results on a computer screen for professionals such as doctors and nurses. Sensor’s data are sent to the same data package in order to identify that the data have been sent via a wireless network from the sensor nodes of each room. The data from each patient are sent to process and display in the healthcare database center as shown in Figure 5 and can be initially classified into three levels: normal, cautious, and risky levels.

Things Board, an open-source IoT platform available for free with a charge for additional functions, is used for the Wireless Communication for both the web server and the data base (cloud). The Things Board is used for classifying data structures, connecting to devices, keeping data security, sending notifications, displaying data, and supporting several protocols such as MQTT, HPTT, and COAP. It is able to store and download historical data as a CSV file. Therefore, the IoT platform using a Things Board is suitable for an affordable smart health system. An IoT platform can be categorized into three parts: (1) data structure classification, (2) data processing, and (3) data display.
Data structure classification manages a large and complex data into categories ranging in order, which consists of an admin, a customer, an asset, and a device. A device, the smallest category, connects and receives data from the MCU, which sends pulse, body temperature, and blood pressure data to the system. To connect with the device, the MCU has to connect with the internet and have compatible communication standards and passwords as the device. An asset is like a box that supports data from devices that can categorize devices and access and modify all of the devices’ information. A customer manages multiple-user access where each user is able to access and modify data from asset, device, and dashboard authorized by customers. An admin is an administrator of the Things Board system and is able to access and modify all data including customer, asset, device, rule engine, and dashboard.Data processing is the part that analyzes data sent from devices where a term of conditions is set for a notification when errors occur. The data are also processed in controllers of the rule engine, a device in Things Board functions. The rule engine processes all data by using a workflow method that is the combinations of block instructions; thus, no coding is required. It includes data verification, notification, online or offline status detection, structure, and relationship modification, exporting data to an external platform and controlling the MCU. The compositions of the rule engine are: (1) Message serves as data management, which is sent to Things Board; (2) Rule Node, which is a function that has a wide range of actions such as editing, calculating, analyzing data into any condition, changing data relationships, and detection device status; (3) Rule Chain, which collects all function relationships such as an input, message, and rule node and combines them into one process. The purpose of the combination is for the convenience of data use, especially in a case when there is more than one process.The data display can adjust the dashboard screen in various actions such as notifications and status detection. Moreover, it can decide whether to display the data only on the device or the asset, which consists of many devices in the network. Widgets is a tool that has a wide range of actions—for example, digital and analog gauges, charts, maps, control GPIO, table, etc.

## 4. Essential Components and Validation Method

There are three main parts of smart device designs: sensor, microcontroller, and display screen. Figure 6a shows the architecture of the overall system where there are two sensors: MAX-30102 [26] (in Figure 6b) used for monitoring blood pressure and heart rate and GY-906 [27] (in Figure 6c), a sensor for monitoring body temperature. The monitored data are sent to the ESP-32 mini controller [28] to process the data according to the coding algorithm before displaying on the screen (in Figure 6d) and storing in the cloud system. All tools are powered by a Li-polymer battery 502530 3.7 v 360 mAh [29].

In order to test the programming process, all devices were preliminarily installed on a board in the laboratory, and the OLED displaying screen used I2C serial communication, which reduces connection space and requires fewer cables. The PCB circuit design for the smart device consists of an ESP-32 Mini32 V2.0.13 board, used as the MCU to process the data from sensors sent to the cloud system, and a 0.96 Inch OLED display screen. USB Socket Female Type-A 4 Pins were used to connect and receive the data from MAX-30102 sensors installed on the patient’s fingertip. GY-906 Infrared Temperature was used to measure temperature around the wrist. Printed circuit boards were designed by the Proteus program as shown in Figure 7, which shows a 4 cm × 4.4 cm two-page circuit board designed in a single sheet to obtain small printed-circuit boards to fit all devices. After all devices are assembled with the printed circuit board, the smart device is ready for the test as shown in Figure 8. The vital sign signals of the proposed smart devices (PRO) were then validated with the standard medical equipment (STD) for each sensor. The accuracy test is analyzed by using JMP 11 with 60 samples.

## 5. Results

Figure 9 presents a comparison of measured results between the proposed smart devices and the standard medical equipment (IOS Smart Watch) for the heart rate. It can be seen that, by using the GY-906 temperature sensor, the result from the proposed device could measure body temperature with high accuracy where the mean difference between STD-HR (the result from the standard medical equipment testing of the heart rate) and PRO-HR (the result from the proposed smart devices testing of the heart rate) is 0.95, only 1.25%. The Standard Deviation (Std Dev) and Standard error (Std err) for the heart rate measurement were observed and presented in Table 1.

Considering the results of blood pressure measurements, Figure 10 shows the comparisons of Systolic blood pressure (SBP) and Diastolic blood pressure (DBP) between the proposed and standard devices. The overlay plot of the SBP and DBP from the proposed device and the standard device (OMRON HEM-7130) are exceptionally aligned, which ensures that the inexpensive MAX-30102 sensor in the proposed device for measuring blood pressure can be confidently used because the difference between the mean SBP and DBP are only 0.22 and 0.11 mmHg from the standard device, respectively. Dev and Std err were also observed as shown in Table 2 for STD-SBP and PRO-SBP, which are 0.57 mmHg and 0.07, respectively, and 0.06 mmHg and 0.01 for the STD-DBP and PRO-DBP, respectively.

The testing results of the body temperature (BT) comparing the proposed device (PRO-BT), a low-cost infrared thermometer, and the standard device (STD-BT), the OMRON infrared thermometer, is shown in Figure 11. It shows a good agreement between the PRO-BT and STD-BT results in each point with the absolute mean difference of only 0.04 °C for body temperature, which is exceptional. The difference values of the mean, Std Dev, and Std err are observed for BT, which are 0.04 °C, −0.06 °C, and −0.001, respectively, as presented in Table 3.

Data monitoring can be displayed on OLED and dashboard screens. The OLED display is connected with the I2C serial of microcontroller as shown in Figure 12. The dashboard screen in the Things Board platform serves as the master to collect and display data on computers and mobile phones. Data from smart devices are sent to the Things Board platform with terms of conditions, and a notification is displayed when errors occur. The data from smart devices are also sent to identify wireless networks from the sensor nodes of each room to process and display in the healthcare database center. Widget’s tool in Things Board performs the actions of digital and analog gauges, charts, maps, control GPIO, and tables in each room as shown in Figure 13. Cost comparison between the proposed device with the Apple Watch Series 5 and Apple Watch Series 6 along with its battery life were presented in Table 4 where the prices of HR and Blood pressure sensors are approximately 10 USD, the body temperature sensor is 5 USD, and the ESP microcontroller is 12 USD.

## 6. Conclusions

An intelligent medical system with low-cost wearable monitoring devices for measuring basic vital signals of admitted patients was proposed with exceptional accuracy from testing experiments of 60-person sample size. Two main parts were developed, the basic low-cost vital signs sensors and the IoT platform, which make the entire monitoring complete and can be applied for various types of medical applications with considerably low expense and installation time, suitable for field hospitals, bedridden patients, palliative care patients, etc.

The vital signs sensor was developed as a wristwatch able to monitor heart rate, blood pressure, and body temperature of a patient and display the results on the wristwatch screen, while also sending real-time results to Things Board, which is an IoT platform for data management purposes that also performs smart notifications when vital sign signals meet set criteria. The testing experiments of the proposed wristwatch showed an acceptable accuracy level compared with standard devices when testing with 60-patient samples with mean errors’ heart rate of 1.22%, systolic blood pressure of 1.39%, diastolic blood pressure of 1.01%, and body temperature of 0.13%. The real-time updating IoT platform works successfully for data storage, notification, and user division. According to testing results with 10 smart devices connected with the platform, the time delay caused by the distance between smart devices and the router is 10 s each round with the longest outdoor distance of 200 m. As there is a short time-delay, it does not affect working ability of the smart system, making the proposed system able to show patient’s status and be functional in emergency situations. Moreover, the OTA system helps to automatically edit code and update software via a Wi-Fi connection. With the proposed system, the patients did not get disturbed during recovering or monitoring time.

## Figures and Tables

**Figure 1 micromachines-12-00918-f001:**
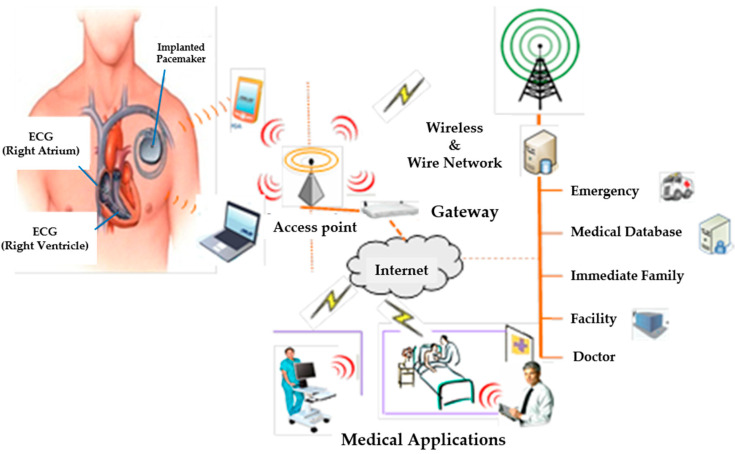
An application of wireless sensor networks for the patients’ body.

**Figure 2 micromachines-12-00918-f002:**
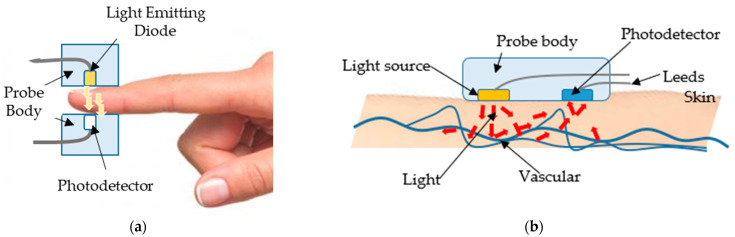
The technique of photoplethysmography (PPG): (**a**) using light transmission property; (**b**) using light reflection property [25].

**Figure 3 micromachines-12-00918-f003:**
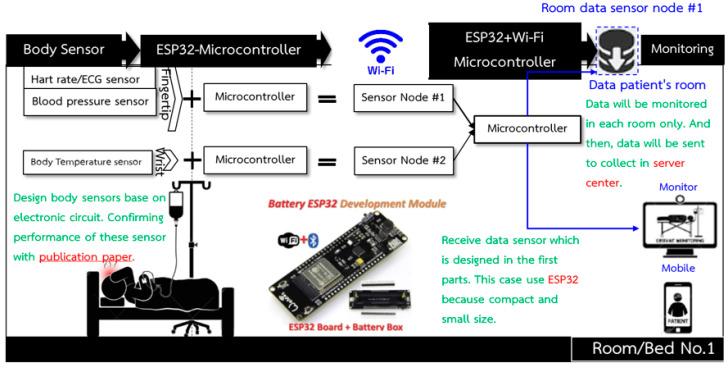
Overall structure of body sensor nodes and the communication pathway.

**Figure 4 micromachines-12-00918-f004:**
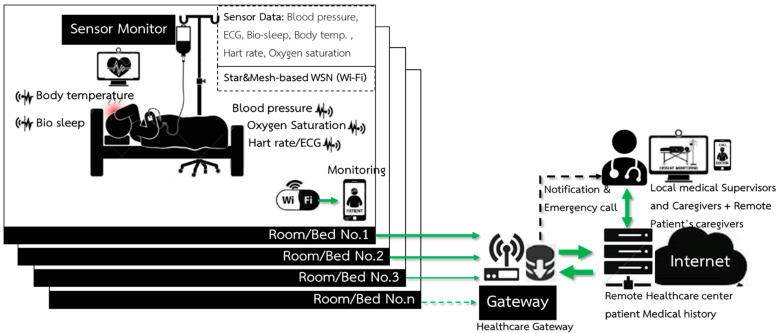
The intelligent wireless network system for monitoring the vital signs of internal patients.

**Figure 5 micromachines-12-00918-f005:**
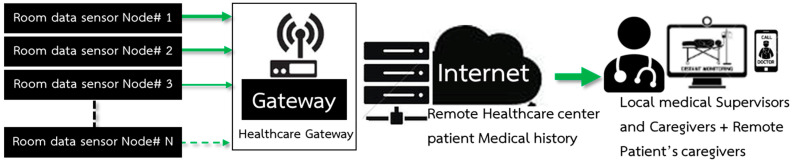
A structure of the communication between sensor node and room data sensor node (#I.D.) to the healthcare database center.

**Figure 6 micromachines-12-00918-f006:**
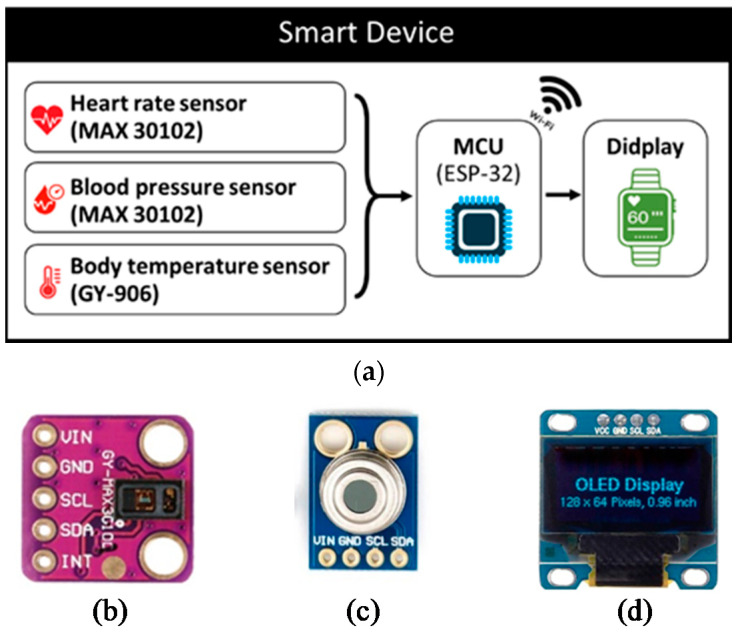
A process flowchart of the device (**a**) tools used in smart device designs; (**b**) MAX-30102; (**c**) GY-906; (**d**) OLED display screen.

**Figure 7 micromachines-12-00918-f007:**
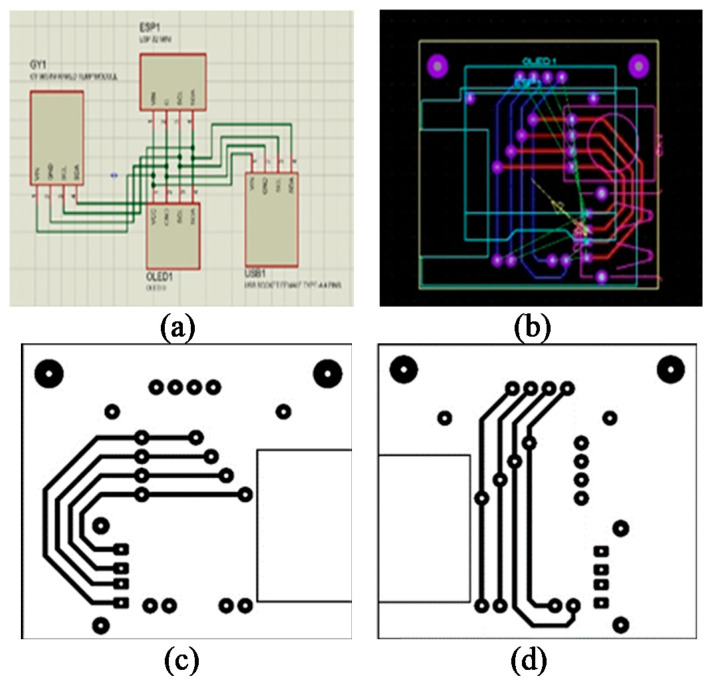
Assembling a smart device (**a**–**d**) the PCB circuit design by the Proteus software.

**Figure 8 micromachines-12-00918-f008:**
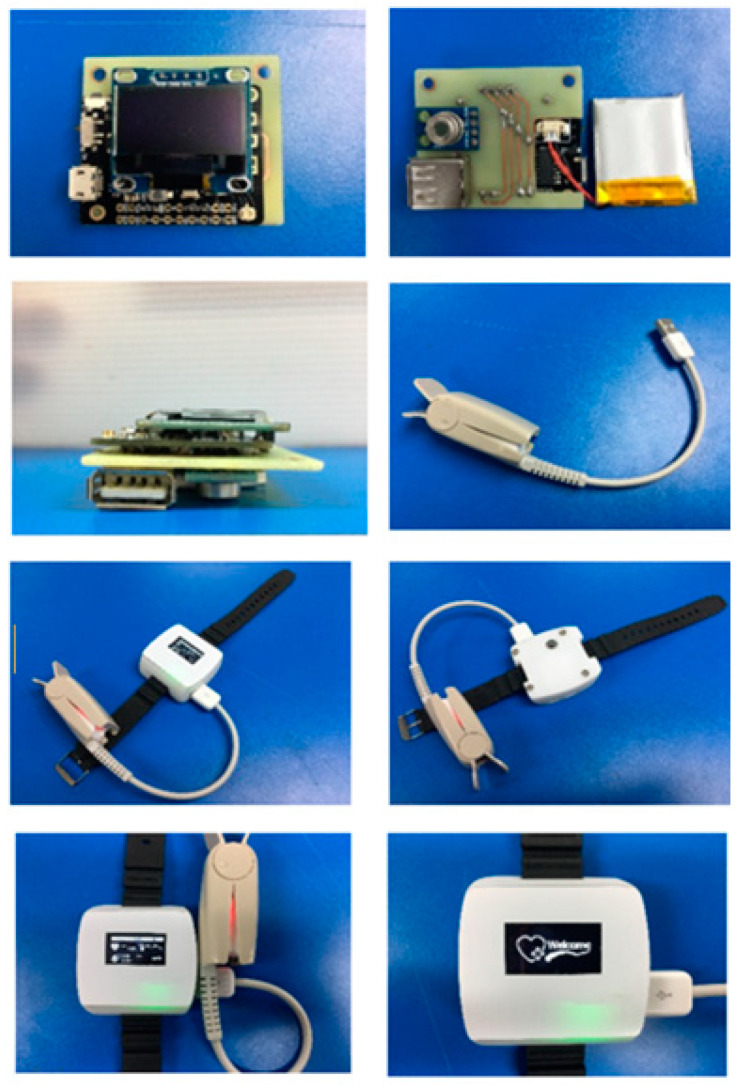
Ready to use sensor devices, MCU and a display screen.

**Figure 9 micromachines-12-00918-f009:**
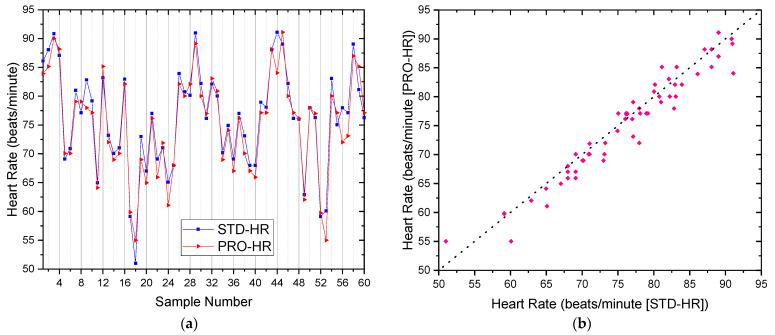
The comparison of measured heart rate results between the proposed smart device and the standard medical equipment (IOS Smart Watch): (**a**) the overlay plot; (**b**) the comparison plot.

**Figure 10 micromachines-12-00918-f010:**
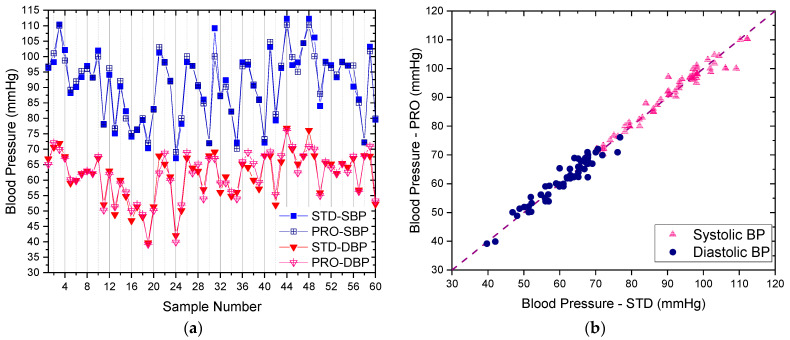
The comparison of systolic and diastolic blood pressure results between the proposed smart devices (PRO) and the standard medical equipment of OMRON HEM-7130 (STD): (**a**) the overlay plot; (**b**) the comparison plot.

**Figure 11 micromachines-12-00918-f011:**
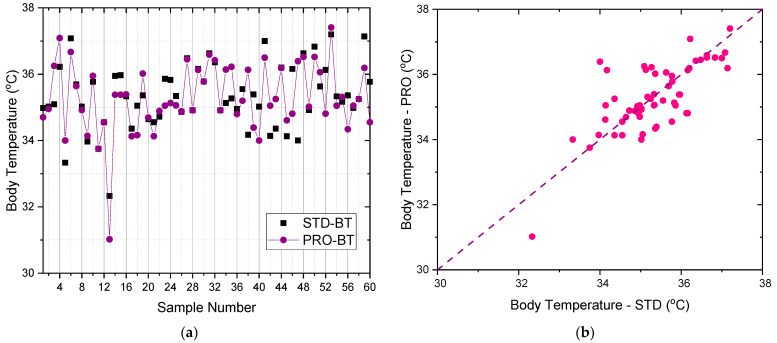
The comparison of body temperature results between the proposed smart devices (PRO) and the standard medical equipment of OMRON infrared thermometer (STD): (**a**) the overlay plot; (**b**) the comparison plot.

**Figure 12 micromachines-12-00918-f012:**
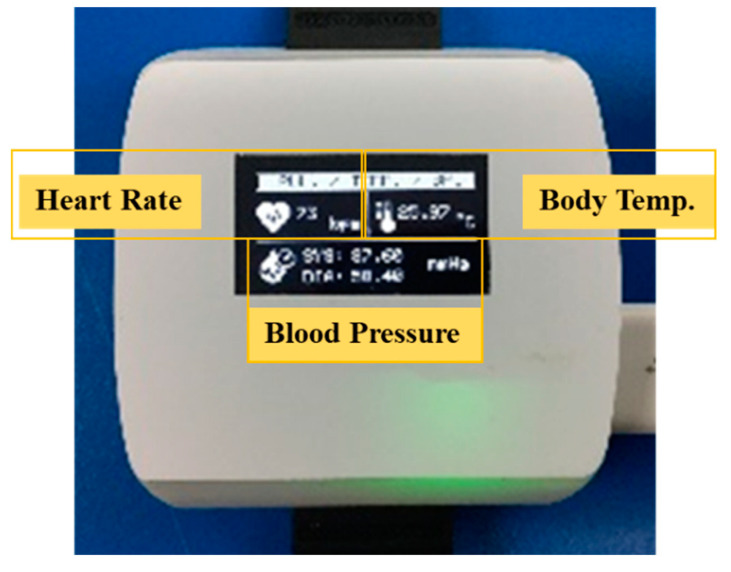
Smart device monitoring on OLED.

**Figure 13 micromachines-12-00918-f013:**
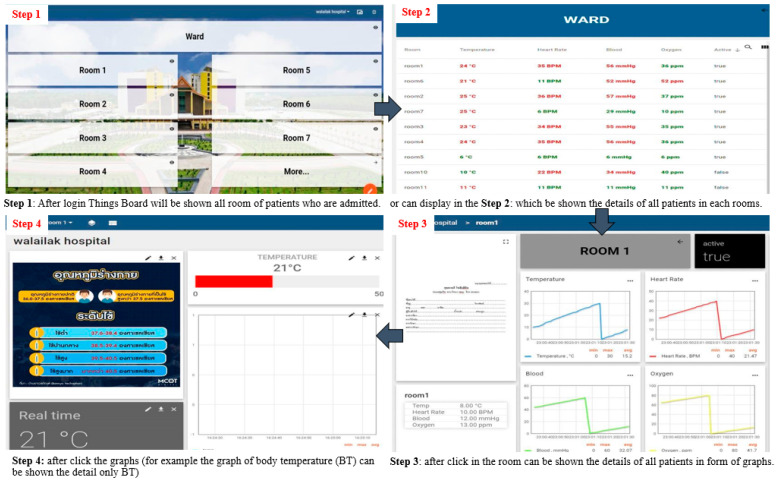
The dashboard screen in the Things Board and steps to display the details of Vital Signals of Patients for nurses and medical professionals monitoring.

**Table 1 micromachines-12-00918-t001:** The descriptive statistics of the heart rate results.

	Mean		Std Dev		Std Err	
N	STD-HR (Beat/m)	PRO-HR (Beat/m)	STD-HR (Beat/m)	PRO-HR (Beat/m)	STD-HR	PRO-HR
60	76.12	75.17	8.73	8.70	1.23	1.12

**Table 2 micromachines-12-00918-t002:** The descriptive statistics of the blood pressure for both systolic and diastolic results.

	Mean				Std Dev				Std Err			
N	STD-SBP (mmHg)	PRO-SBP (mmHg)	STD-DBP (mmHg)	PRO-DBP (mmHg)	STD-SBP (mmHg)	PRO-SBP (mmHg)	STD-DBP (mmHg)	PRO-DBP (mmHg)	STD-SBP	PRO-SBP	STD-DBP	PRO-DBP
60	90.63	90.85	61.37	61.48	11.35	10.77	7.88	7.82	1.46	1.39	1.02	1.01

**Table 3 micromachines-12-00918-t003:** The descriptive statistics of body temperature results.

	Mean		Std Dev		Std Err	
N	STD-BT (°C)	PRO-BT (°C)	STD-BT (°C)	PRO-BT (°C)	STD-BT	PRO-BT
60	35.31	35.27	0.96	1.03	0.12	0.13

**Table 4 micromachines-12-00918-t004:** Cost and battery life comparisons between commercial and proposed smart wearable devices.

Healthcare Features	Apple Watch Series 6	Apple Watch Series 5	Proposed Watch
Size (mm)	44/40	44/40	60
Heart Rate	✓	✓	✓
Body Temperature	x	x	✓
Blood Pressure	x	x	✓
SpO_2_	✓	x	x
Charging	Wireless	Wireless	USB
Battery Life	18 h	18 h	15 h
Prices (USD)	Start +400	Start +400	30

## Data Availability

Not Applicable.

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
