# Peer review of "Intelligent Medical System with Low-Cost Wearable Monitoring Devices to Measure Basic Vital Signals of Admitted Patients"

_micromachines, 2021, doi:10.3390/mi12080918_

Round 1
Reviewer 1 Report
The authors present a system to measure basic vital signals of patients. Vital signs sensors have been developed together with an IoT platform.
In the system architecture, it is reported that every sensor is connected to a microcontroller that send the data via Bluetooth to another microcontroller connected via wifi to a cloud. Why did you not send the data directly via wifi to the cloud for subsequent processing? The ESP32 microcontroller is equipped with both WiFi and Bluetooth
Did you evaluate the energy consumption and in particular the battery lifetime?
The related works should be analyzed in more detail, other similar works are reported in the literature, some of them are cited here after, please explain the difference between your system and those already published.
K. C. Chowdary, K. Lokesh Krishna, K. L. Prasad and K. Thejesh, "An Efficient Wireless Health Monitoring System," 2018 2nd International Conference on I-SMAC (IoT in Social, Mobile, Analytics and Cloud) (I-SMAC)I-SMAC (IoT in Social, Mobile, Analytics and Cloud) (I-SMAC), 2018 2nd International Conference on, 2018, pp. 373-377, doi: 10.1109/I-SMAC.2018.8653716.
P. S. Akram, M. Ramesha., S. A. S. Valiveti, S. Sohail and K. T. S. S. Rao, "IoT based Remote Patient Health Monitoring system," 2021 7th International Conference on Advanced Computing and Communication Systems (ICACCS), 2021, pp. 1519-1524, doi: 10.1109/ICACCS51430.2021.9441874.
S. Marathe, D. Zeeshan, T. Thomas and S. Vidhya, "A Wireless Patient Monitoring System using Integrated ECG module, Pulse Oximeter, Blood Pressure and Temperature Sensor," 2019 International Conference on Vision Towards Emerging Trends in Communication and Networking (ViTECoN), 2019, pp. 1-4, doi: 10.1109/ViTECoN.2019.8899541.
Lei Ru et al. "A detailed Research on Human Health Monitoring System Based on Inernet of Things" https://doi.org/10.1155/2021/5592454
Wan, J., A. A. H. Al-awlaqi, M., Li, M. et al. Wearable IoT enabled real-time health monitoring system. J Wireless Com Network 2018, 298 (2018). https://doi.org/10.1186/s13638-018-1308-x
Nizar Al Bassam, Shaik Asif Hussain, Ammar Al Qaraghuli, Jibreal Khan, E.P. Sumesh, Vidhya Lavanya, IoT based wearable device to monitor the signs of quarantined remote patients of COVID-19, Informatics in Medicine Unlocked,
Volume 24, 2021, 100588, ISSN 2352-9148, https://doi.org/10.1016/j.imu.2021.100588.
Moreover, similar devices are available, for example:
https://caretakermedical.net/
You claimed that the developed system is low-cost compared to what is already on the market, but without an overall assessment of costs how can you assert this?
minor mistakes:
row 142: In the fourth section (add In)
Author Response
The authors are grateful for the Editor and reviewers’ comments. The Authors carefully made corrections and modifications in the manuscript (m/s), which are presented in track change version. Line numbers are included in our detailed answers in the attached file.

Reviewer 2 Report
It is proposed to add, new, proposal 'phase sub-figures', next to the plots/diagrams of the Figure 9 (and the Figures 10,11). The new 'phase-diagram sub-figures' might implement another, more semantic, 'X:axis' (than "No. of Samples"). These new, proposal, 'figures' might have as X-Y the comparable measurements, e.g. "X:-> measurements of the standard medical equipment"
and "Y:-> measurements of the proposed smart device".
Also there are some (9) notes/corrections:
1. Line104: A mistyping error ('. withal'), in: "data can be processed and display to patients. withal processed data can also be sent to a".
2. Line333: Add a reference for specifications of the 'MAX-30102', in: "are two sensors which are MAX-30102 (in Figure 6(b)) used for monitoring blood pressure".
3. Line335: Add a reference for (the 'ESP-32') specifications, in: "from body area. The monitored data is sent to the ESP-32 mini controller to process the 335".
4. Line337: Add a reference with the battery specifications, in: "and storing in the cloud system. All tools are powered by a Li-polymer battery 502530 337
3.7v 360mAh.".
5. Line345: Add a reference with the specifications of the 'GY-906', in: "30102 sensors installed in the patient's fingertip. GY-906 Infrared Temperature was used".
6. Line372: A mistyping error. Reoplace 'results' by 'heart rate results', in: "Figure 9. The overlay plot platform to compare measured results between the proposed smart".
7. Line380: Explain 'what are these sensors'. Also add a reference per sensor, in: "sure that the inexpensive sensors used in the proposed device can be confidently used as".
8. Line386: A mistyping error. Reoplace 'results' by 'blood pressure both systolic and diastolic results results', in: "Figure 10. The Overlay Plot platform to compare the results between the proposed of the smart...".
9. Line389: Add the missing units of the blood pressure (for the Mean, and for the Std Dev), as well as on the 'Y:axis' of the Figure 10, in Table 2 (and the blood pressures in: Line383, Line384), in: "Table 2. The descriptive statistics of the blood pressure both systolic and diastolic results.".
10. Line389: Add the missing units of the body temperature (for the Mean, and for the Std Dev), as well as on the 'Y:axis' of the Figure 11, in: "Table 3. The descriptive statistics of body temperature results.".
Author Response

(The authors gave the same response as above.)

Round 2
Reviewer 1 Report
the authors have answered exhaustively to my comments, the article can be accepted in its current form